# Perceptions, experiences and barriers to lifestyle modifications in first-generation Middle Eastern immigrants to Sweden: a qualitative study

Patricia Olaya-Contreras,[1] Katarina Balcker-Lundgren,[2] Faiza Siddiqui,[2,3] Louise Bennet[2,3]

[1]Institute of Health and Care Sciences, Sahlgrenska Academy, University of Gothenburg, Gothenborg, Sweden
[2]Center for Primary Health Care Research, Region Skåne and Lund University, Malmö, Sweden
[3]Department of Clinical Sciences, Lund University, Malmö, Sweden

**Correspondence to**
Dr Patricia Olaya-Contreras;
patricia.olaya-contreras@gu.se

## ABSTRACT

**Objective** The prevalence of type 2 diabetes (T2D) among Iraqi immigrants to Sweden is high and partly related to sedentary physical activity and calorie dense food. The aim of the present study was to explore perceptions, experiences and barriers concerning lifestyle modifications (LSM) in Iraqi immigrants to Sweden at risk for T2D.

**Design** A qualitative thematic analysis was conducted on data collected from gender-specific focus group interviews which took place during a culturally adapted randomised controlled intervention study addressing motivation to lifestyle change, self-empowerment, behavioural modifications and sociocultural barriers to LSM. Seven focus groups were held, with an interval of 1–4 weeks between January and May of 2015; each session lasted approximately 1.5 hours.

**Setting** The city of Malmö, Sweden.

**Participants** Out of 27 women and 23 men assigned to the intervention group, 19 women and 14 men who attended at least one focus group session were included in the study.

**Results** Participants expressed awareness of the content of healthy lifestyle practices. They also expressed numerous social and cultural barriers to LSM connected to irregular meals, overeating, food and drinking preferences and family expectations. Overeating was described as a consequence of social and cultural norms and expectations and of poor mental well-being. Facilitators for reaching successful LSM were connected to family involvement and support.

**Conclusion** Our study reports that facilitators for LSM are connected to presence of family support. Preventive actions addressing family involvement may benefit Middle Eastern immigrants at high risk for T2D to consider healthier lifestyles practices. Identification of sociocultural barriers and facilitators for LSM are crucial for successful health promotion in minority populations at risk for T2D.

**Trial registration** Trial registration number: NCT01420198 for the MEDIM-study; Pre-results.

## BACKGROUND

Political instability in the Middle East during recent decades has forced millions of people to flee their homelands, and a large proportion of these migrants subsequently reach Europe. Today, first-generation immigrants from the Middle East constitute approximately 4% of the Swedish population, with the largest groups born in Iraq or Syria.[1]

Western versus non-western cultures often differ in terms of religion, traditions and social norms.[2] According to the WHO, health is governed by so called 'social determinants' including environmental, socioeconomic factors and cultural norms influencing lifestyles.[3] Poor socioeconomic conditions contribute to higher diabetes incidence. For instance, non-western immigrants including Middle Eastern immigrants from Iraq to Denmark are often referred to live in socioeconomically deprived areas thus leading to higher morbidity and mortality rates.[4] In Sweden, refugees assigned to live in neighbourhoods of high deprivation developed type 2 diabetes (T2D) to a higher extent compared with those assigned to a neighbourhood of low deprivation.[5] Structural and sociocultural norms in immigrants and refugees influence lifestyle habits, that is, calorie

dense foods and physical inactivity and are correlated with a high risk for T2D.[6–8]

► A strength of this study is the use of focus group discussions, which encourage interaction between respondents to explore and clarify individual and shared perspectives on lifestyle modifications (LSM) and views on the intervention.
► To address language barriers, a native Arabic speaker was present in all sessions who performed back and forward translations when transcriptions were done.
► Perspectives on sociocultural barriers to LSM of younger, second-generation Iraqi immigrant women and men are not represented in this study.
► Men had a lower attendance rate, and barriers to LSM may be under-reported.

In Sweden, the prevalence of T2D among Middle Eastern immigrants is almost twice as high as compared with that of the native population.[9 10] Despite more healthcare visits and the fact that they receive pharmacological treatment at an earlier stage, non-western immigrants have worse metabolic control, and develop diabetes complications at an earlier stage, compared with native Swedes.[11]

In ethnic minority populations, lifestyle interventions have been shown to be unsuccessful, and individuals at risk do not reach target unless interventions are culturally adapted.[7] In Europe, the few randomised culturally adapted interventions conducted on lifestyle mainly focused on South-East Asian immigrants, such as the Inva-Diab Deplan study and minority health study among Pakistani women,[12] the Physical activity and minority health study on Pakistani men in Norway, the Podosa trial in the UK[13] and the SAHELI study on Surinamese immigrants in the Netherlands.[14] However, religion, traditions and governing social norms as well as living context differs between South Asian and Middle Eastern populations and also differs to the countries they have migrated to. Hence, this study was conducted to investigate the influence of sociocultural factors on lifestyle practices among Middle Eastern immigrants, specifically Iraqi immigrants, to increase the understanding of barriers and facilitators to lifestyle modifications (LSM) that Iraqi immigrants to Sweden have experienced.

Most qualitative studies have focused on studying the effect of sociocultural factors on LSM in ethnic minority populations from South-East Asia to Europe and the USA. The aim of the present study was accordingly to explore perceptions, experiences and barriers to LSM in immigrants from Iraq with overweight and at high risk of developing T2D.

## METHODS
The present study used a qualitative research design in the collection and analysis of data based on a culturally adapted intervention, the MEDIM, impact of Migration and Ethnicity on Diabetes in Malmö cohort study 2010–2012.[9]

### Recruitment of participants
In the MEDIM study, approximately 45% out of the 1398 Iraqi-born participants were identified as being at high risk for developing T2D during the next decade by being obese (body mass index (BMI) >28 kg/m$^2$) or having prediabetes. In 2015, these at risk individuals (n=636) fulfilled inclusion criteria and were invited to participate in this randomised controlled trial (RCT) of 4-month duration consisting of a culturally adapted LSM programme that aimed to reduce diabetes risk.[15] Those diagnosed with diabetes, pregnant and/or poor physical or mental health were excluded from the study. A total of 96 first-generation Iraqi born immigrants signed informed consent to participate. Half of the participants were randomised to the intervention group (n=50) and half to the control group (n=46). None of the participants were known to the investigators prior to participation. The information analysed in this study was gathered from the intervention group and the discussions from the group sessions.

### Procedure and data collection
The intervention group were offered to participate in a total of seven gender-specific group sessions, which were conducted in their native language. The sessions addressed awareness, motivation, taking action, relapse, review and feedback, setting goals and self-empowerment. The content of each group session is presented in table 1, and was based on the evidence-based guidelines for prevention of T2D.[16] In accordance with the expected content of culturally adapted interventions,[6 7] the sessions were 'culturally adapted', that is, addressed knowledge gaps, gender-specific groups and sociocultural barriers to lifestyle change.[15] Participants were also provided economic support for gym classes and gym clothing. The interview guide for the focus group sessions included structured and open-ended questions to explore a specific set of topics; first by asking a broader question addressing difficulties in reaching the goals and subsequently asking more specific questions (table 1). Participants individually answered the questions and were encouraged to interact with each other to further explore the views on the intervention. The topics in the interview guide are presented in detail in the online supplementary file 1. The interview guides were pilot tested prior to the intervention. An interdisciplinary team (specialist nurse in diabetes, health coach and Arabic and Swedish speaking translator) conducted the sessions. Each session took approximately 1.5 hours, with intervals of 1–4 weeks over a duration of 4 months.

All sessions were recorded. The interview recordings were subsequently translated by the official translator from Arabic to Swedish and transcribed verbatim in Swedish. To check the translation from Arabic to Swedish, transcripts were checked against the recordings for accuracy, checking with the translator grammar and reviewing the translated data transcripts and notes for data analyses by two of the authors (KB-L, LB) who are native Swedes. Thereafter, three of the authors (KB-L, LB, PO-C) were responsible for checking the accuracy of the translation of the cited quotations from Swedish to English. This ensured accuracy of the translation of the cited quotations, that is, confirming credibility.

### Data analysis
In analysing the data gathered from the focus group interviews, an inductive thematic analysis was conducted, that is, that the codes and themes derived from the content of the data themselves.[17] This method is an independent and reliable qualitative approach to analysis that provides

**Table 1** Content of the discussions at each focus group session

| | |
|---|---|
| Session 1 (awareness) | The group discussion addressed how unhealthy lifestyle habits predict risk of diabetes and cardiovascular disease, as well as discussions on nutritional contents of different foods. |
| Session 2 (motivation/getting started) | It focused on understanding the participant's current dietary and physical activity habits. The action plan was introduced as a tool that helped to formulate goals for lifestyle change and to support motivation for implementing lifestyle change, that is, to write the personal goal and describing how a participant would reach it. |
| Session 3 (taking Action/maintaining the change) | It was designed to get started by putting dietary change into an action plan, by stimulating motivation for dietary change and PA and further by discussing how to achieve the postulated goals. The session also included discussions on common cooking methods for Iraqi food and how to make dishes healthier, that is, consisting of less fat, sugar and more fibres. |
| Session 4 (cooking session) | In the cooking session, participants were encouraged to bring the recipe of their favourite dish. This recipe was then modified by a professional chef at the cooking session to include healthier dishes including more fibres, less fat as well as less carbohydrates in the diet. The goal of this session was to highlight how cooking method, food preparation and portion size were vital components for effective lifestyle change. |
| Session 5 (relapse—maintaining the change) | To follow-up the action plan, and to identify barriers to its implementation, as well as experiences regarding social support and family engagement towards the intervention. Facilitated rewrote action plan. |
| Session 6 (review and feedback) | Participants had documented their everyday PA in a diary and this session included discussions on the final outcomes of lifestyle change, importance of maintaining lifestyle change, motivation to set new goals and discussions on external social support/network to reach those goals. The session also included feedback from the participants. |
| Session 7 (setting goals and self-empowerment) | This session highlighted the importance of regularly setting new goals and reviewing these goals. This session also focused on empowering the participants and addressed how they had handled barriers and how they had evolved and reached their goals throughout the process. |

PA, physical activity.

a purely qualitative, detailed and nuanced account of the data.[17 18] The data were organised to extract emergent patterns in the semantic content. All the authors read the transcribed data multiple times. A thematic code book was developed to facilitate identification of main themes, thus, allowing for the multiple coverage of possible topics. The first author made codes by marking important features in the transcribed text systematically across the entire data set and later collating data relevant to each code. Thereafter, the codes were organised into related areas to identify patterns and to construct descriptive themes. Then, a thematic map to make clear definitions and names for each theme was made. The authors met to discuss the codes to identify new themes potentially converging or diverging, and further, to explore possible additional meanings in the content of participants' narratives.[17] This ensured that the complexity of experiences was reflected in the analysis (ie, dependability, confirmability, credibility). A saturation point was attained when there was no additional information in terms of non-emergence of new codes of themes (inductive thematic saturation).[19] A model exemplifying the analysis process, showing meanings and interpretations of theme B is provided in table 2. Major themes were defined as those responses/opinions voiced by more than half of the participants. NVivo software program V.10 was used in the coding process.

### Ethical considerations
All participants provided voluntary written informed consent prior to their participation in the present study. They were informed about the possibility to withdraw from the study without explanation and without consequences. All data collected were anonymised and stored on a protected server only accessible by selected members of the research team.

### Patient and public involvement statement
The participants were not involved in the design or planning of the study. The findings will be disseminated to the public through the present scientific publication in *BMJ Open*, interviews, lectures and media.

### RESULTS
A total of 19 women and 14 men who participated in at least one group session were included in this study (table 3). The majority had not completed secondary school (86%). Almost every second participant was unemployed (52%). Most were married with a family size of two to six persons. Female participants were, on average, 8 years older than female non-participants (50.7 vs 42.6, p=0.049), whereas there were no significant age differences between male participants and non-participants (48.0 vs 46.4 years, p=0.747). There were no significant differences in BMI for male and female participants versus non-participants (males 31.4 kg/m$^2$ vs 30.7 kg/m$^2$ p=0.699; females 30.8 kg/m$^2$ vs 31.2 kg/m$^2$ p=0.833). The main reason for non-participation was not receiving permission from work. Figure 1 presents participation rate in relation to total number of sessions during the

**Table 2** The analysis process, of meanings and interpretations of theme B 'perceptions, experiences and barriers towards healthier lifestyles

| Theme | Subtheme | Meanings and interpretations |
|---|---|---|
| Perceptions, experiences and further barriers for healthier lifestyles | Challenges with regular meals | ► Loneliness at home affects willingness to eat regularly. <br> ► Eating fastfood instead of regular meals. <br> ► Not perceiving the need for regular mealtimes and unestablished mealtime schedules. <br> ► Eating repeatedly snacks and sweets. <br> ► Sometimes overeating or fasting. |
| | Food preparation, taste preferences and portion size | ► Preferences for eating food with 'good smell and taste' over dishes prepared in a healthier way (boiled). <br> ► Difficulties to introduce to the diet healthier food, healthier cooking and low-calorie products. <br> ► Custom of eating large portions. |
| | Family traditions and expectations | ► The children decide what to eat in the family hindering a healthy diet. <br> ► Following family traditions when gathering to eat hindering the willingness to eat moderately. <br> ► Family resistance to exercise. <br> ► Challenges changing the role of the mother; limiting the mother's time and possibilities to exercise. |
| | Other barriers influencing motivation level | ► Climate and related factors lessening outdoor physical activity and increasing overeating. <br> ► Depressed mood isolation affecting willingness to participate in outdoor physical activities. <br> ► Difficulty to establish an exercise routine after many years of physical inactivity that worsens with concomitant pain. |

intervention. Fifty per cent of male participants participated in only one session; however, 14% attended almost all sessions. Women had a more even attendance rate with 10%–20% attending at least four sessions.

The findings are condensed into three major themes: (aA) Motivation to lifestyle modifications: awareness of healthy diet habits and physical activities; (B). Perceptions, experiences and barriers towards healthier lifestyles and (C) Experienced obstacles and facilitators for maintaining lifestyle modifications following the intervention.

**Table 3** Group sessions with number of female and male participants throughout the intervention

| Session | Women | Men | Total/session |
|---|---|---|---|
| Session 1: awareness | 14 | 9 | 23 |
| Session 2: motivation/getting started | 15 | 7 | 22 |
| Session 3: taking action/maintaining the change | 14 | 4 | 18 |
| Session 4: cooking session | 9 | 7 | 16 |
| Session 5: relapse—maintaining the change | 9 | 3 | 12 |
| Session 6: review and feedback | 9 | 4 | 13 |
| Session 7: setting goals and self-empowerment | 7 | 3 | 10 |

### Motivation to lifestyle modifications: awareness of healthy diet habits and physical activities (sessions one and two)

The expression 'healthy lifestyles' were understood as being physically active, that is, doing sport/exercise, going to the gym regularly, eating vegetables and fruits, fibre-rich and balanced diet:

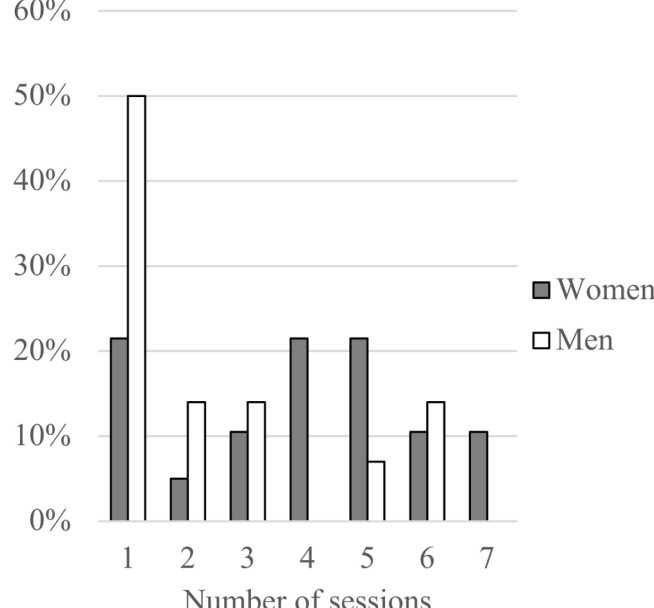

**Figure 1** Gender participation rate in relation to total number of the sessions during the intervention.

To avoid sweets such as cookies, to drink water; To eat vegetables, as much as possible; To be physically active at least half an hour every day; To walk often; To reduce weight; You should sleep at least 7 hours; You should eat cooked food; One should not eat fried food; You should drink a lot of water and eat fish; You should eat regularly.

They also expressed awareness of unhealthy lifestyles:

To consume too much red meat; to be obese or overweight; to drink soft drinks; to eat candies; to get stressed.

Physical inactivity was associated with illness and disease:

The reason why I have been physically active is to lose weight; I do not want a stroke or diabetes; Many in my family suffer from diabetes, so, I do not want to suffer from it.

### Perceptions, experiences and barriers towards healthier lifestyles (sessions three to five)
#### The challenge with regular meals
The participants described that they had difficulties having regular meals such as breakfast, lunch and dinner, often skipping the main meals, and commonly replacing meals with sweets. Among them, it was also common to have a late dinner, that is, at night, sometimes they overate or on the contrary ate almost nothing.

Some participants explained that the reason was because their children ate at school, and when they felt hungry they provided themselves with some snacks or fast food:

I do not usually eat so much lunch, I eat some chocolate when I get tired in the afternoon; I eat when I feel hungry, I do not think so much about what time to eat; I eat many meals often; At home, sometimes I go walking and eat, I have no specific time to eat.

#### Food preparation, taste preferences and portion size
Fried or sweet tasting foods were preferred and perceived as 'food with good smell and taste', which they described as highly appreciated:

You do not feel good if you do not eat fried foods; I believe that I eat more fried products after walking because I need it.

Some participants did not like the taste of water; instead they drank sodas and/or sweet juices. Others disliked the taste of foods boiled in water instead of being fried. A common statement was they neither consumed integral products with fibre-rich flour nor low-calorie products. The ways to prepare foods or the kind of ingredients used were perceived as the most difficult to change:

I found it very difficult to change our diet, from the food that we like to eat, but I thought, I would be able to consume less sweet products

Participants liked to eat large portions. They had a habit of serving large portions, eating fried snacks, sweet beverages (sodas and tea with sugar) and candies. Some women believed that they were overweight because of how much they had eaten in each meal and how much food they had eaten between meals rather than because of how the food was prepared:

I realize that the problem, for me, it is the amount of food and the portion size.

#### Family traditions and expectations
A common statement was that the children decided what and where to eat, which explained some of the difficulties to change eating habits. The mothers were eating fast food as their children did because of 'the need to follow family decisions' and, in addition, the women explained their overeating because 'the boys need to eat enormous portions after exercising in the evening'. As a result of this, they prepared a lot of food and therefore they were serving themselves with big portions.

Nevertheless, some participants mentioned that after eating in excess they became sad or upset with themselves. Some participants drank soda on a daily basis (at least four cans per day) but were concerned with this and expressed that it needed to be stopped.

Family traditions and expectations influenced eating habits when celebrating or meeting relatives:

Yes, when I meet relatives, they invite me [to eat] all the time, and I cannot say no…; this is considered impolite not to eat the food when offered; It is hard to say no to good food that you get served; My family thinks I'm too small, but I'm overweight! They offer me different dishes and meals the whole time.

Some of the women expressed interest in exercising, but they sometimes felt alone in this effort and did not always feel support from their family members for exercising. They also described they did not have enough free time for outdoors physical activity or for going to the gym. Women commonly described that their time was mostly dedicated to household chores and time with the children:

I like to exercise but it's hard to find time when I have children; in my family, my children and my husband are not engaged in exercise and thus, I will be alone in my effort to exercise; I live in Sweden, I do not work, I take care of the household, children, shopping and cooking and have no time for sports; It's difficult to get time for exercise, but I try to be physically active as often as it is possible.

#### Other barriers influencing motivation level
The participants expressed that they struggled to adapt to the cold and dark climate, which had a negative influence on outdoors physical activity, as one of the participants illustrated:

I get depressed when I eat less…; The weather is boring and the darkness too and I do not have so many social contacts; When I cook and the kids do not eat that much, I begin eating instead of exercising

It was stated that after some years without exercise, it was difficult to establish a routine and feel motivated to exercise regularly, and this feeling was worse when suffering from pain. One woman explained:

I have lived in Sweden for 8 years, I have four children, and I have not exercised in these years.

A male participant expressed:

Even though I'm on sick-leave, preferred to do other activities at home and do not have time to exercise.

### Experienced obstacles and facilitators for maintaining lifestyle modifications following the intervention (session 6–7)
#### The paradox: raised awareness but still no lifestyle modifications
Despite the raised consciousness concerning the health benefits from changing dietary habits and increasing level of physical activity after the intervention—especially the women—described that they continued eating sweets and pastries mainly when they were alone, and copiously, large portions when sharing with family and relatives. They found it difficult to lose weight, which they experienced, was related to social/family pressure. Most participants considered the goal of 10 000 steps/day to be demanding because of lack of time to exercise and not feeling comfortable to go for a walk alone.

In fact, some of the women disliked go to the gym, they considered the gym as a 'stuffy place';
Further, they stated it was difficult to be alone there.

We would like to go to places only for women where we can exercise and do sports. These facilities need to be located near where we live; otherwise, it is not possible for us to go there.

#### Changing lifestyle habits together
In some of the families, spouses were involved and supported the modifications implemented by their partner. In other families, the sessions influenced the whole family, that is, controlling their children's intake of sugar and or preparing some dishes in a healthier way:

The whole family has changed dietary habits; My husband now eats only hard rye bread now and no white bread anymore; We eat a lot of vegetables, not like before; We bought whole grain rice and started using it instead; My husband has begun to check calories and ingredients in the food he buys, and now he is aware of the food we should not consume and those we should.

Some participants described that they took regular walks together with family members, some started going to the gym:

I have started to walk a lot and be more physically active; I have started with water aerobics three times a week and working out twice a week; I also walk; I'm doing gymnastics, training 1 hour each day, 3 times a week and really, I feel better inside and outside; I had tried the treadmill and have trained with the machines in the gym. It is necessary to be strong to be able to lose weight and to get a healthier life.

Two of the women involved their family members in domestic tasks, facilitating their possibility of having some extra time for physical activity:

My daughter and I walk together but my steps are more than hers are. I'm walking fast, I usually walk from 11 o'clock to 17; My son is practicing every day with me in the gym, so he helps me; I have lost 5–6 kilos, I think.

During the intervention, some participants lost weight after increasing their everyday physical activity, despite some health problems, which is illustrated here:

Before, I weighed 77 kilos, and now I weigh 70 kilos; My goal was weight loss and although I have pain in my knee, I lost 10 kg; I have walked, and I need to continue walking or exercising to lose weight; I have focused on physical training and if I cannot walk, I exercise at home or go swimming.

## DISCUSSION
### Key findings
This is one of the first qualitative studies investigating self-perceived experiences, barriers and facilitators to LSM in Middle Eastern immigrants to Europe, representing a growing population at high risk for T2D. In this study of first-generation Iraqi immigrants to Sweden, family expectations and structural norms were identified as both strong barriers and facilitators for maintaining healthy lifestyle practices. Culturally adapted lifestyle intervention studies have been conducted previously in Europe; however, these studies are few and address the South Asian population.[12–14 20] This study is novel in that it addresses the Middle Eastern immigrant population.

We have previously shown that this culturally adapted RCT study significantly reduced cardiometabolic risk in the intervention as compared with the control group. Further, this study showed positive impact of the intervention and group sessions on reported LSM and on objectively assessed physical activity as well as metabolism.[15 21]

### Lifestyle challenges in the Middle Eastern immigrant population
The previous population based MEDIM study found a considerable proportion of the Iraqi immigrant population at very high risk for developing T2D within a decade.[9] The Middle Eastern population have other cultural aspects and traditions (such as language, food, religion and sociocultural norms) as compared with the South

Asian population. Consequently, this RCT intervention study was initiated as a health initiative to prevent the Middle Eastern population from developing cardiometabolic disease; this study successfully reduced the cardiometabolic risk in these participants by combining different strategies.[15]

Different theoretical models link socioeconomic and cultural factors with poor health. For instance, in the 'Transitional Model', Schlossberg concludes that sets of factors related to contextual and individual factors as well as environmental support seem to affect how one copes with transition and adaption to a new lifestyle.[22] Neighbourhood deprivation predicts incidence of diabetes in refugees in Sweden,[5] which is in agreement with the WHO, stating that social determinants of health such as socioeconomic conditions, culture and social norms impacts health in chronic diseases morbidity.[3] Hence, LSM are fruitless unless environmental, social and cultural barriers are identified and addressed.[5 7] Furthermore, the term 'social capital' has been suggested to be associated with health through several different pathways, including psychosocial mechanisms, norms regarding health-related behaviours and access to healthcare, factors that at the population level are important for the maintenance of health and reduced mortality rates.[23] Middle Eastern immigrants to Sweden in general and Middle Eastern women, in particular, are exposed to all these different levels of social determinants, which is reflected by the fact that >60% of first-generation Iraqi immigrants in Malmö report low social capital (assessed as low social participation, low social anchorage, low instrumental and low emotional support)[24] as well as high rates of morbidity in non-communicable diseases and poor mental health.[25] The low participation in this study could be a reflection of poor social capital, with low societal trust.

### Perceptions, experiences and barriers towards healthier lifestyles
Although the awareness of the impact of healthy lifestyles on health, participants experienced difficulties to modify their lifestyles. The participants perceived that over time it was difficult to establish new dietary habits and become more physically active, which is consistent with studies showing that individuals living in socioeconomically vulnerable areas, challenges in achieving modifications are often hindered by contextual and family influences in achieving LSM.[26 27]

The group sessions addressed knowledge gaps regarding how lifestyle influences the risk of developing T2D. However, the participants perceived as difficult to modify dietary preferences, especially when sharing with family and friends. The perceptions that food boiled in water rather than stir fried or unsweetened drinks do not taste good is a challenge to conquer, especially since eating patterns are commonly guided by family and children's expectations which in accordance with prior research.[26 28–30]

Women commonly described feeling alone in their efforts to change their lifestyles. They felt that family expectations such as taking care of the children and managing the household had a negative influence on their ability to be physically active, which is consistent with what has been reported in other studies.[6–8 25–27 29 31 32]

In the intervention, 10 000 steps per day was set as a daily goal, but it was perceived by most participants, as being unreachable. Beginning with lower step goals can enhance adherence to the intervention in sedentary groups.[33] Thus, in this study, a reason for not complying with the intervention could be due to the perception that the goals were too difficult to reach.

Iraqi immigrants in Sweden and Pakistani immigrants in Norway show high prevalence of poor mental health and a strong connection between anxiety and depression with physical inactivity.[25 34] Iraqi born participants in this study experienced that to cope with the dark cold climate and depressed mood, they overate and preferred snacks instead of regular meals, which is in line with prior studies conducted among immigrant women at risk for T2D.[26 27 29 31]

### Facilitators for maintaining lifestyle modifications
Despite living in Sweden for more than two decades over 70% of Iraqi born immigrants in Malmö do not read or write in Swedish, and prefer traditional calorie dense foods,[35] reflecting low level of integration and acculturation.[24] Family traditions may impose cultural and social norms on each individual, being a barrier for successful LSM. On the other hand, our data indicate that strong family support being a cornerstone in the Middle Eastern culture represents an important health-promoting factor. The Transition Model highlights the importance of receiving support from both, family and environment to enhance adaptation, that is, to cope with the intervention. Individuals move through the transition process, through a series of phases, that is, 'moving in', 'moving through' and 'moving out',[22] which may apply to our participants who need to cope, adapt and make decisions across some cultural and social situations to succeed with LSM.

The participants who managed to change their lifestyle described a profound support and involvement of their family members in this effort, that is, motivating each other to maintain healthier habits. These findings correspond with previous research, in which, involving family in target interventions promotes healthier eating behaviours, increased physical activity as well as well-being among immigrant populations.[6 26 27 29 30 36]

The availability for exercise centres, arenas and other physical environments are important contributors to lifestyles changes.[7 26 30 37] In Malmö, there are several gyms for women and in this study, 'Hälsans Hus', that is, gym facilities only for women located in the nearby neighbourhood, was one of the physical activity centres women were referred to. Nevertheless, the perceived lack of nearby gender-specific gym facilities was a common reason among women to continue being physically inactive. Lack of time was another reason for not performing regular physical activity in this study. A previous study conducted

in Sweden showed that Middle Eastern women wanted non-strenuous 'suitable and pleasant' exercise, that is, cycling, swimming and gymnastics and that it is important that the exercise is 'proper', that is, not to exercise in front of the men.[32] Hence, close collaboration with the communities and families is mandatory to fully understand and adjust physical activity for different needs among non-westernised female immigrants, as confirmed by previous research.[7 29 32 33 37]

### Strengths and limitations of this study

Strengths of the study were the group discussions encouraging respondents to explore individual and shared perspectives and views on the intervention and focus attention on possibilities to improve healthy lifestyles. A native Arabic speaker was present in all sessions and led back and forward translations when transcriptions were performed, providing accuracy and credibility of the transcribed and analysed information. In the last session—session #7, the participants provided feedback and addressed how they had handled obstacles and reached their goals throughout the intervention, thus, assuring trustworthiness and confirmability of the information collected in this study. The participants did not know each other prior to the participation. Couples who had participated in the RCT study were randomised to either the intervention or control group,[15] thus, avoiding spread of information between the groups (in this qualitative study, only the intervention group was included).

Some participants might have feared to express themselves openly in the group sessions. According to culturally adapted interventions,[7] this issue was addressed by having gender-specific groups to assure participants as far as possible could feel free to express their opinions. Although men participated to a lower extent than women, both men and women contributed to all themes as illustrated in figure 1 where sessions 1 and two represented the theme 'Motivation to lifestyle change'; sessions 3–5, 'Perceptions, experiences and barriers towards healthier lifestyles'; sessions 6 and 7, 'Experienced obstacles and facilitators for maintaining lifestyle modifications following the intervention'. In accordance with the Ethics statement, participants could leave the study without any explanation, thus, we do not have data on all reasons for leaving; however, a main reason for not participating was not receiving permission from work.

Our findings are limited to middle-aged first-generation Iraqi born immigrants rather than younger second-generation participants. Further, ethnicity includes a variety of religious beliefs and family history; therefore, caution needs to be made when interpreting the data and when transferring our findings to other ethnical minority groups. However, the participants in this study have previously been considered representative for the Iraqi immigrant population living in Sweden.[9] Moreover, the findings of barriers to lifestyle change reported here correspond with previous culturally adapted intervention studies of other non-western immigrant populations as referred to above.

## CONCLUSIONS

Identification of sociocultural barriers and facilitators for LSM in ethnic minority populations at increased risk of T2D are crucial for both healthcare professionals and society to successfully support populations at risk for T2D and enable them to successfully reach healthier lifestyle practices. The present study highlighted that facilitators for LSM were connected to profound family support. Tailoring interventions to include entire families, that is, possibilities for childcare and the involvement of spouse/other significant family or community member in the intervention could be successful strategies for increasing adherence to interventions/programmes among Middle Eastern immigrants.

Barriers and facilitators for lifestyle change in ethnic minority populations are crucial for healthcare and society to acknowledge in the continued effort to promote healthier LSM in vulnerable populations at risk for non-communicable diseases.

**Acknowledgements** We wish to acknowledge the invaluable contribution of the study participants. We are grateful to health coach Muna Mohammed, translator Iman Yousif, study Nurses Saranda Muhaxheri, Josefin Goode Khan and Asma Saleh, administrator Sonja Ruhnke and Ida Hagman. We are also grateful to Hälsans Hus, Rosengård, Abena, Viktväktarna, Arbetarnas Bildningsförbund (ABF) and Oatly. We thank Josephine TV Greenbrook for her help with the English language in the initial draft of the manuscript.

**Contributors** PO-C: Wrote and revised the manuscript, analysed and interpreted the data, translated the material coming from the group discussions from Swedish to English. KB-L: Participated in conducting the group sessions; participated in data acquisition, reviewing the translated data transcripts and notes from Sweden to English for data analyses and data assessment; contributed with interpretation of the data and participated in writing and revising the manuscript. FS: Participated in the design of the study and in conducting the group sessions; participated in assessing data and interpretation of the data interpretation and contributed to writing the manuscript. LB: Designed and conducted the original study, contributed data acquisition, reviewing the translated data transcripts and notes from Sweden to English for data analyses and interpretation, data assessment and interpretation of the data; contributed to writing and revised the manuscript.

**Funding** This work was supported by grants from Lund University (ALF grants Dnr 20101641, 20101837 and 162641), Region Skåne (Dnr 226661 and 121811), the Swedish Society of Medicine (Dnr 97081 and 176831), the Crafoord Foundation (Dnr 20110719), the Swedish Research Council Linné grant to LUDC (Dnr 349-2006-237 and Dnr 349-2008-6589), Strategic Research area Exodiab (Dnr 2009-1039), Swedish Foundation for Strategic Research (Dnr IRC15-0067) and ANDIS (Dnr 825-2010-5983).

**Competing interests** None declared.

**Patient consent for publication** Not required.

**Ethics approval** The Ethical Review Board of Lund University, Sweden approved the study (approval no. 2011/88).

**Provenance and peer review** Not commissioned; externally peer reviewed.

**Data availability statement** Data can be made available upon request from the corresponding author.

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
