## [Reviewer comments · BMJ Open]

ARTICLE DETAILS

TITLE (PROVISIONAL)	Perceptions, experiences and barriers to lifestyle modifications in first-generation Middle Eastern immigrants to Sweden – A qualitative study
AUTHORS	Olaya-Contreras, Patricia; Balcker-Lundgren, Katarina; Siddiqui, Faiza; Bennet, Louise

VERSION 1 – REVIEW

REVIEWER	Laura Terragni Oslo Metropolitan University (OsloMet)
REVIEW RETURNED	23-Feb-2019

GENERAL COMMENTS	Firstly, I'd like to apologize to the authors and the editor for the delayed review. New job commitment had made my last month more busy than expected. I know by being an author myself how frustrating it is to wait! This is even more important given my critical review to the article. I hope the editor would communicate to the authors the reasons of my suggestion of rejecting the article and that this can contribute to rework with your article. Following are some of the main reasons that lead me to this decision. I read the article several times, it is well written and potentially interesting. However I did not find much that could increase our knowledge about barriers to lifestyle changes among immigrant population and about the need of cultural adaptation. The reasons that "no other study on Middle East immigrant have been conducted before" is not sufficient. You need also to show what IS SPECIFIC about this group and how studying the challenges of this group can contribute to the large existing literature of culturally adapted intervention in Europe. Lack of references to some important studies on culturally adapted interventions in Europe is a limitation. Another limitation is the lack of description of the cultural adaptation of the intervention and its justification. Methodologically there are some contradictions as the authors stated firstly that they did a deductive thematic analysis (line 23 p.9) but then talk about "emergent themes" (l.32 same page) which is more appropriate for an inductive approach. How the themes emerged is not described. Most participant did not participated to all the sessions of the interventions this limitation should have been discussed properly. However, what I consider the main limitation of the study is a kind of "taken for granted" culture differences between "us" and "them" and a superficial approach to culture differences. This is particularly evident in lines 36-45 page 7. Firstly the statement is very generic, secondary by stating that in non-western population there are sociocultural barriers to lifestyle changes, it seems to imply that such barriers are not present in western population. Besides this very rough simplification between western and not
---

	western (unfortunately common in the literature), it is important to recognize that lifestyle changes are difficult for all. Of course than there are, differences: but are these differences only due to culture or are other aspects important as well? What about the intersection between migration, acculturation and social class? In an eventual resubmission to this or other journals, I really recommend to have a more reflexive attitude towards cultural and social differences. Good luck with your work!
--	---

REVIEWER	Dr. Nitha Mathew Joseph Cizik School of Nursing at UTHealth The University of Texas Health Science Center at Houston USA
REVIEW RETURNED	06-Apr-2019

GENERAL COMMENTS	Overall the manuscript is well written. Missing some components. 1. The Significance of the study or the need for the study is missing in the abstract. 2. Specifically write the name of population as Iraqi immigrants in line 19 under background section as each non -European - immigrant population's lifestyle is different and may not be able to develop preventive interventions for then based on this study results. 3. Explain the term non-westernized in line 4 background. 4. In response to the line 43-45;read the below studies (attached link) conducted among minority groups https://www.ncbi.nlm.nih.gov/pmc/articles/PMC5705764/ https://www.ncbi.nlm.nih.gov/pmc/articles/PMC4666510/ 5. Add a detailed description of demographic characteristics of the sample in the result section and represent it in a table format also. 6. Along with eligibility criteria write both inclusion and exclusion criteria of selection of study participants. 7. Change the word "patients " to participants in line 25.
---

REVIEWER	Juliet Aweko Karolinska Institutet
REVIEW RETURNED	07-Apr-2019

GENERAL COMMENTS	This is an interesting and definitely relevant topic to explore. Currently as the manuscript is, there is need for major revisions to enhance credibility and trustworthiness of the results. I have a few major concerns highlighted below. 1) In the introduction page 6 lines 55 and page 4 lines 7, you mention that a few studies have focused on studying the effect of cultural and social factors on LSM but you don't mention the studies or their findings for that matter. What is the current state of LSM in your study populations or similar setting, how have the findings from the previous studies informed your research question? 2) In the methods section you also don't explicitly describe the characteristics of your study population or refer the reader to any table with that information. - You mention that the study participants were part of an intervention but do not clearly describe the intervention, what did the intervention include? - The topics in the interview guide are not specific, perhaps provide samples of the interview questions and attach the guides as supplementary material for review
---

	- There is no table or model showing the analysis process - how did you move from the codes to the major themes, this makes it difficult for the reader to trust the process and outcome of the analysis 3) You mention that the focus group discussions included men and women but the results provide only the women's perspective, why is that so? what happened to the men? - Additionally the quotes provided to support the text in the respective themes do not give detail of which interview number they were extracted from and which participants. Its makes me question the validity of the results - Moreover the text in each of the themes is too limited in detail, not enough to justify the themes - What is the difference between categories 3.3.1 and 3.3.2? - You may need to revisit the analysis and show the different levels of abstraction 4) I miss the key points for discussion in the discussion section - In the introduction you mention that there are few studies exploring your topic area but in the discussion but suddenly, there are several studies concurring with your findings, how is that? - This is a qualitative study but there is no discussion on credibility and trustworthiness of your study. Did you attain saturation? please provide more detail of this. In conclusion, I don't think your results currently support your conclusions. There is need for further analysis in-order to arrive to your conclusions.
--	---

REVIEWER	Ashley Robertson Coventry University, UK
REVIEW RETURNED	10-May-2019

GENERAL COMMENTS	See attached file I was asked to provide a review with a particular emphasis on the methods and analyses. I would like to point out that the research topic is not in my area of expertise; therefore my comments around the Introduction and Discussion are limited. Introduction  • Para 1, Sentence 2: Something is off with the structure of this sentence. Methods  • I see that the researchers did not know the participants; however, did participants know each other in the focus groups? • Unsure about the inclusion of 'Dissemination of the findings' – it doesn't add anything in my opinion • A little more detail specifically about how your method was modified from Braun and Clarke's method would be useful. Results  • I think the themes appear to be well-described and evidenced • I don't think it's necessary to shorten physical activity to PA; at times it flits between the two. Discussion  • I don't know the literature, but seems fine. • Linked to theory, which is good.
--

	General  • Ensure the tenses are correct (e.g. that past is used most of the time). Some (non-exhaustive) examples:  o ‘...96 consent’ o ‘Some testimonies illustrate...’ • There is a need for further proof-reading across the manuscript, to ensure the grammar and punctuation is sound. Some examples that need revised:  o ‘They all origin from the Middle East’ (‘originate’) o “Of them, 96 consent...’ (‘Of those, 96 consented) o “less than the secondary school” o “make other activities at home” o “a reason for not complied” • Generally, there are too many acronyms used. I’d advise trying to only use them for the most important things (e.g. T2D etc).
--	--

VERSION 1 – AUTHOR RESPONSE

Reviewer: 1

Reviewer Name: Laura Terragni

Institution and Country: Oslo Metropolitan University (OsloMet) Please state any competing interests or state ‘None declared’: None declared

Please leave your comments for the authors below

Firstly, I’d like to apologize to the authors and the editor for the delayed review. New job commitment had made my last month more busy than expected. I know by being an author myself how frustrating it is to wait! This is even more important given my critical review to the article. I hope the editor would communicate to the authors the reasons of my suggestion of rejecting the article and that this can contribute to rework with your article. Following are some of the main reasons that lead me to this decision.

1. I read the article several times, it is well written and potentially interesting. However I did not find much that could increase our knowledge about barriers to lifestyle changes among immigrant population and about the need of cultural adaptation. The reasons that “no other study on Middle East immigrant have been conducted before” is not sufficient. You need also to show what IS SPECIFIC about this group and how studying the challenges of this group can contribute to the large existing literature of culturally adapted intervention in Europe. Lack of references to some important studies on culturally adapted interventions in Europe is a limitation.

Reply: We agree that culturally adapted lifestyle intervention studies have been conducted in immigrant populations to western countries. However, it is not only that these studies are few rather than numerous, but they also address the South Asian population rather than the Middle Eastern immigrant population. As previously reported lifestyle interventions are fruitless unless culturally adapted and since governing cultural habits, social norms and traditions differ (such as language food, religion and social norms) between Middle Eastern and Asian populations we have chosen to address the Middle Eastern immigrant population since they –rather than the Asian population– represent the largest non-European immigrant population in Sweden today, that are at high risk for type 2 diabetes. The issue raised by the reviewer is now addressed in the Background (p.6-7) as well as in paragraph in the Discussion section (p.17-18, 22). The reference list is updated.

2. Another limitation is the lack of description of the cultural adaptation of the intervention and its justification.

Reply: In the Methods section, p.8 under the subtitle Procedures, we have now clarified the justification as well as content of the cultural adaptation of the intervention.

3. Methodologically there are some contradictions as the authors stated firstly that they did a deductive thematic analysis (line 23 p.9) but then talk about “emergent themes” (l.32 same page) which is more appropriate for an inductive approach. How the themes emerged is not described.

Reply: Thank you for your remark. An inductive thematic analysis was done, it means that the codes were created inductively to capture the meaning, and content of each sentence. The information is now edited, p.9.

4. How the themes emerged is not described.

Reply: Thank you for a valuable comment. The analysis process, in the Methods section under Data analysis p 9-10 was edited and now described in more detail, explaining how the themes emerged, i.e., showing how we moved from the codes to the major themes. Additionally, we have added Table 2, with a model for Theme B exemplifying the different levels of abstraction in this theme. Furthermore, the Results section is edited accordingly.

5. Most participant did not participated to all the sessions of the interventions this limitation should have been discussed properly.

Reply: Thank you for this comment. We have now discussed that in the Discussion, section p 19 (last paragraph), p 21, and under limitations, p 22-23.

6. However, what I consider the main limitation of the study is a kind of “taken for granted” culture differences between “us” and “them” and a superficial approach to culture differences. This is particularly evident in lines 36-45 page 7. Firstly the statement is very generic, secondary by stating that in non –western population there are sociocultural barriers to lifestyle changes, it seems to imply that such barriers are not present in western population.

Reply: Thank you for your remark. The impression you refer to above, does not represent our values or intentions. The section is now rephrased and the whole manuscript edited accordingly now discussing cultures differences (Background p.6, p.7, and Discussion p 18-19) in a more nuanced way including a discussion on Transferability (Strengths and Limitations p 22-23).

7. Besides this very rough simplification between western and not western (unfortunately common in the literature), it is important to recognize that lifestyle changes are difficult for all. Of course than there are, differences: but are these differences only due to culture or are other aspects important as well? What about the intersection between migration, acculturation and social class? In an eventual resubmission to this or other journals, I really recommend to have a more reflexive attitude towards cultural and social differences. Good luck with your work!

Reply: We agree that the intersection between migration, acculturation and socioeconomic factors impacts morbidity and mortality in cardiometabolic diseases; as mentioned above have now addressed and discussed that more thoroughly in the Background as well as in the Discussion section. The whole Discussion section has also been edited to present a more clear Discussion with subheadings addressing each topic. The reference list is updated.

Reviewer: 2

Reviewer Name: Dr. Nitha Mathew Joseph Institution and Country: Cizik School of Nursing at UTHealth, The University of Texas Health Science Center at Houston, USA Please state any competing interests or state ‘None declared’: none

Please leave your comments for the authors below Overall the manuscript is well written. Missing some components.

1. The Significance of the study or the need for the study is missing in the abstract.

Reply: The abstract is updated accordingly.

2. Specifically write the name of population as Iraqi immigrants in line 19 under background section as each non-European-immigrant population's lifestyle is different and may not be able to develop preventive interventions for then based on this study results.

Reply: The sentence is updated now and Transferability discussed under Strengths and Limitations p 22.

3. Explain the term non-westernized in line 4 background.

Reply: An explanation is now added to the introduction and this is supported by a reference.

4. In response to the line 43-45; read the below studies (attached link) conducted among minority groups <https://www.ncbi.nlm.nih.gov/pmc/articles/PMC5705764/>

<https://www.ncbi.nlm.nih.gov/pmc/articles/PMC4666510/>

Reply: Thank you for valuable references. We have updated the reference list in the manuscript now.

5. Add a detailed description of demographic characteristics of the sample in the result section and represent it in a table format also.

Reply: Characteristics of the study participants are now described in the Results section, p.10.

6. Along with eligibility criteria write both inclusion and exclusion criteria of selection of study participants.

Reply: This information is now given in the Methods section, under Participants: Inclusion criteria (obesity or prediabetes) as well as exclusion criteria are included in the Methods section under the paragraph Participants p.8.

7. Change the word "patients " to participants in line 25.

Reply: This is rephrased.

Reviewer: 3

Reviewer Name: Juliet Aweko

Institution and Country: Karolinska Institutet Please state any competing interests or state 'None declared': Non

Please leave your comments for the authors below This is an interesting and definitely relevant topic to explore. Currently as the manuscript is, there is need for major revisions to enhance credibility and trustworthiness of the results. I have a few major concerns highlighted below.

1) In the introduction page 6 lines 55 and page 4 lines 7, you mention that a few studies have focused on studying the effect of cultural and social factors on LSM but you don't mention the studies or their findings for that matter. What is the current state of LSM in your study populations or similar setting, how have the findings from the previous studies informed your research question?

Reply: The reference list is updated now including an updated list of current LSM publications in similar settings, i.e. Europe, and a brief presentation of these studies is now written in the Introduction. We have removed the line on page 4 line 7 since this is not a main finding but rather belongs to future studies and perspectives. In a paragraph added to the Discussion section we discuss how our previous research has influenced our data p. 18.

2) In the methods section you also don't explicitly describe the characteristics of your study population or refer the reader to any table with that information.

Reply: Thank you for a valuable comment. We have now added a paragraph in the Results section describing the characteristics of our study population, p.10-11.

- You mention that the study participants were part of an intervention but do not clearly describe the intervention, what did the intervention include?

Reply: The methods section is now updated to clarify this. We have clarified the content of the cultural adaptation of the intervention and referred to the original paper where the whole content of the Intervention is described in detail p 7-8.

- The topics in the interview guide are not specific, perhaps provide samples of the interview questions and attach the guides as supplementary material for review

Reply: The topics discussed are previously presented in Table 1 and the topics including the Interview guide used in the sessions are now included as Supplementary data.

- There is no table or model showing the analysis process - how did you move from the codes to the major themes, this makes it difficult for the reader to trust the process and outcome of the analysis

Reply: Thank you for this feed-back. The analysis process is in the Methods section under Data analysis p. 9-10 described in more detail, explaining the construction of the Themes and how the themes emerged, i.e., showing how we moved from the codes to the major themes. Additionally, we have added Table 2, with a model for Theme B exemplifying the different levels of abstraction in this theme. Furthermore, the Results section is edited accordingly.

3) You mention that the focus group discussions included men and women but the results provide only the women's perspective, why is that so? what happened to the men?

Reply: This is a misinterpretation. As illustrated in Figure 1, men also participated in the group sessions however to a lower extent than women. In the results section, we present expressions made by both men and women however, some comments are specifically expressed by women and some by men and we have now clarified that in the text. The reasons for not participating in the interviews are mentioned in the results section and discussed under limitations.

- Additionally the quotes provided to support the text in the respective themes do not give detail of which interview number they were extracted from and which participants. Its makes me question the validity of the results.

Reply: In the Results section we now describe which group session the themes are derived from. We have edited the themes in the Results section so that they now present which sessions they correspond with. We have anonymized the identity of the participants, and organized each participant contribution by session to ensure the contribution of each participant to the respective group discussion. According to the ethical agreement, data should not be identifiable and trackable to specific participants. Thus the information is not presented in relation to which participant said what. Further, we have clarified in the Results section the content of the subjects discussed in each group session that are presented in Table 1. Number of participants in each group session is presented in Table 3 and further, figure 1 shows proportion of participants contributing to the discussions in each session.

- Moreover the text in each of the themes is too limited in detail, not enough to justify the themes

Reply: Thank you for a valuable comment. The Results section is edited accordingly with both, the analysis and the content of the sessions, presenting the different levels of abstractions for each theme. The analysis process was revisited and it is now described in more detail the Methods section under Data analysis p 9-10, and explains how the themes emerged.

Moreover, to increase the details and content around the group sessions and themes the subtitles describing each theme in the Results section now describe what group session they are derived from. Further, the topics discussed in the interviews are now presented in Table 1 as well as in a new table (Supplementary data).

.

- What is the difference between categories 3.3.1 and 3.3.2?

Reply: Thank you for a valuable observation. We agree with you the difference is small and have removed the subtitle 3.3.2, and have now edited the Theme and Sub-themes in line with the content of each session.

- You may need to revisit the analysis and show the different levels of abstraction

Reply: The analysis process was revisited and it is in the Methods section under Data analysis p 9-10 now described in more detail, explaining the construction of the Themes and how the themes

emerged, i.e., showing how we moved from the codes to the major themes. Additionally, we have added Table 2, with a model for Theme B exemplifying the different levels of abstraction in this theme.

4) I miss the key points for discussion in the discussion section

Reply: Thank you for a valuable comment. The Key points are now clarified in the Discussion section.

- In the introduction you mention that there are few studies exploring your topic area but in the discussion but suddenly, there are several studies concurring with your findings, how is that?

Reply: We have clarified that we refer to the fact that few LSM studies are conducted in Middle Eastern populations. In the Background as well as Discussion, we discuss our results in relation to findings in other studies conducted on minority populations and the reference list is updated.

- This is a qualitative study but there is no discussion on credibility and trustworthiness of your study. Did you attain saturation? please provide more detail of this.

Reply: Regarding credibility and trustworthiness of the study, this is now addressed in the Methods section p 8-10, as well as in the Discussion section p 22-23. In this study, saturation was attained and considers saturation as the point at which 'additional data do not lead to any new emergent codes or themes', thus, saturation focuses on "the identification of new codes or themes, and is based on the number of such codes or themes rather than the completeness of existing theoretical categories" (Saunders et al., 2018). This information is added in the Methods section, and also referred properly. In the Methods section, clarifications of the procedure and analysis of the data is now explained more thoroughly and how this contributed to credibility and trustworthiness. Under 'Strengths and limitations' we have now added a paragraph thoroughly discussing limitations regarding participation in the study, cultural aspects influencing participation as well as methodological topics that may affect our results in terms of credibility and trustworthiness. Transferability is also discussed now.

In conclusion, I don't think your results currently support your conclusions. There is need for further analysis in-order to arrive to your conclusions.

Reply: We have now edited the whole manuscript and have addressed all points raised. We think this has increase the clarity of our results and conclusions but not least improved the description of the analysis process reflecting credibility, confirmability and trustworthiness of our data.

Reviewer: 4

Reviewer Name: Ashley Robertson

Institution and Country: Coventry University, UK Please state any competing interests or state 'None declared': None declared

Please leave your comments for the authors below.

I was asked to provide a review with a particular emphasis on the methods and analyses. I would like to point out that the research topic is not in my area of expertise; therefore my comments around the Introduction and Discussion are limited.

Introduction

Para 1, Sentence 2: Something is off with the structure of this sentence.

Reply: Thank you, we agree. The sentence is now edited.

Methods

- I see that the researchers did not know the participants; however, did participants know each other in the focus groups?

Reply: Thank you for your remark. Participants connections with each other or the research team are now presented in the Methods section p8 as well as discussed under strengths and limitations p22-23. Briefly the participants did not know each other. This information is written in p.22. There were couples participating in the study and they were in the initial study randomized to participate in either intervention or control group to avoid contamination between intervention and control groups. In this study, only the intervention group participated, and men and women participated in separate groups

to increase the probability they did not influence each other in their opinions regarding the subjects discussed in the sessions.

- Unsure about the inclusion of ‘Dissemination of the findings’ – it doesn’t add anything in my opinion
- o Reply: We agree, this is not relevant in this context. However, it is a request from the Editorial board of the Journal.
- A little more detail specifically about how your method was modified from Braun and Clarke’s method would be useful.
- o Reply: Thank you for a valuable observation. The Methods section and Data analysis is now edited to increase the clarity over the process applied in this study.

Results

- I think the themes appear to be well-described and evidenced
- I don’t think it’s necessary to shorten physical activity to PA; at times it flits between the two.

Discussion

- I don’t know the literature, but seems fine.
- Linked to theory, which is good.

General

- Ensure the tenses are correct (e.g. that past is used most of the time). Some (non-exhaustive) examples: o ‘...96 consent’ o ‘Some testimonies illustrate...’
- o Reply: We agree and the text is edited and proofread by a native English speaking medical editor.
- There is a need for further proof-reading across the manuscript, to ensure the grammar and punctuation is sound. Some examples that need revised:
 - o ‘They all origin from the Middle East’ (‘originate’)
 - o “Of them, 96 consent...’ (‘Of those, 96 consented)
 - o “less than the secondary school”
 - o “make other activities at home”
 - o “a reason for not complied”
- o Reply: We agree and the text is edited and proofread by a native English medical writer.
- Generally, there are too many acronyms used. I’d advise trying to only use them for the most important things (e.g. T2D etc).
- o Reply: We will consider that in the manuscript, however we have kept the acronym LSM since that is frequently used.

VERSION 2 – REVIEW

REVIEWER	Nitha Mathew Joseph, PhD, RN Cizik School of Nursing at UTHealth The University of Texas Health Science Center at Houston, Texas USA
REVIEW RETURNED	12-Jul-2019

GENERAL COMMENTS	Thank you for making the recommended changes to manuscript to enhance scientific value. Great job!
--